# Biological Age Predictors: The Status Quo and Future Trends

**DOI:** 10.3390/ijms232315103

**Published:** 2022-12-01

**Authors:** Veronika V. Erema, Anna Y. Yakovchik, Daria A. Kashtanova, Zanda V. Bochkaeva, Mikhail V. Ivanov, Dmitry V. Sosin, Lorena R. Matkava, Vladimir S. Yudin, Valentin V. Makarov, Anton A. Keskinov, Sergey A. Kraevoy, Sergey M. Yudin

**Affiliations:** Federal State Budgetary Institution, Centre for Strategic Planning and Management of Biomedical Health Risks, Federal Medical Biological Agency, Pogodinskaya Str. 10 Bld. 1, Moscow 119121, Russia

**Keywords:** biological age, molecular clock, age-related diseases, life expectancy, COVID-19

## Abstract

There is no single universal biomarker yet to estimate overall health status and longevity prospects. Moreover, a consensual approach to the very concept of aging and the means of its assessment are yet to be developed. Markers of aging could facilitate effective health control, more accurate life expectancy estimates, and improved health and quality of life. Clinicians routinely use several indicators that could be biomarkers of aging. Duly validated in a large cohort, models based on a combination of these markers could provide a highly accurate assessment of biological age and the pace of aging. Biological aging is a complex characteristic of chronological age (usually), health-to-age concordance, and medically estimated life expectancy. This study is a review of the most promising techniques that could soon be used in routine clinical practice. Two main selection criteria were applied: a sufficient sample size and reliability based on validation. The selected biological age calculators were grouped according to the type of biomarker used: (1) standard clinical and laboratory markers; (2) molecular markers; and (3) epigenetic markers. The most accurate were the calculators, which factored in a variety of biomarkers. Despite their demonstrated effectiveness, most of them require further improvement and cannot yet be considered for use in standard clinical practice. To illustrate their clinical application, we reviewed their use during the COVID-19 pandemic.

## 1. Introduction

Age is a major risk factor for chronic noncommunicable diseases, such as heart disease [1], cancer [2], chronic obstructive pulmonary disease [3], Alzheimer’s disease [4], etc. It is a recognized contributor to severe COVID-19 and associated complications [5]. However, many studies have suggested that it is biological rather than chronological age that underlies the development of numerous diseases. People age at a different pace, which is determined not only by genetic predisposition but also by external factors, such as socioeconomic factors and lifestyle. The likelihood of aging-associated diseases and mortality varies even among people of the same age; hence, it could be reflective of their biological age.

The last 15 years saw the emergence of various biological age markers. Ideally, they should correlate with chronological age and be predictive of age-related diseases and mortality. Clinicians use several tests as markers of biological age: maximal oxygen consumption, forced expiratory volume in 1 s, vertical jump, grip strength, whole-body reaction time, unilateral distance, sit-and-reach test, systolic blood pressure, waist circumference, and soft lean mass [6]. Certain inflammatory markers have also been associated with age: IL-6, IL-8, IL-15, IL-1β, TNFα [7,8,9], lipid profile (HDL cholesterol, LDL cholesterol, triglycerides [7,8,10,11]), glucose metabolism profile (glycohemoglobin (Hba1c) and glucose (fasted or oral glucose tolerance test (OGTT) [12]), insulin and C-peptide [13]. Kidney function indicators, such as creatinine, cystatin C, urea, and albumin, have also been associated with age [14]. Microbiome analysis is another way of assessing biological age, since the microbiome has been significantly associated with age [15].

Aging leads to increased genome instability, which can be evaluated using micronucleus assay [16]. Age has also been associated with telomere length [17] and an increase in reactive oxygen species [18]. However, the most common marker of biological age is DNA methylation. It is widely used in forensic medicine as the most reliable age estimator. Other age-associated epigenetic markers could be changes in miRNA concentrations [19], histone modifications [20], and chromatin remodeling [18]. Biological age predictors mentioned in this review are presented in Figure 1.

Individually, these markers are not informative due to their non-specificity. Moreover, changes in their levels can be a manifestation of age-associated conditions, rather than an indication of age. These markers are effective estimators in large study cohorts; however, they may vary significantly at the individual level in clinical practice [21]. To overcome these limitations, artificial intelligence has been used to create models that consider a variety of factors. These models are widely used in clinical practice. They can predict mortality from all causes and the incidence of major aging-associated diseases, including hypertension, diabetes, cardio-vascular diseases, stroke, cancer, and dementia [22,23].

## 2. Biological Age Predictors

### 2.1. Clinical Parameters and Blood Biochemistry as Markers of Aging

#### Blood Biochemistry-Based Calculators

Individually, most clinical biomarkers are insufficiently sensitive to measure the pace of aging and biological age. Studies, however, have shown that certain combinations of biomarkers are more reliable predictors of biological age or mortality. Table 1 presents the main characteristics of the blood biochemistry-based calculators.

Putin E. et al. developed the first blood marker-based model of aging using a group of 21 deep neural networks (DNNs) that were trained on more than 60,000 samples from common blood biochemistry and cell count tests [10]. For each patient, they used only 41 biomarkers; nonetheless, the DNN group achieved a rather small interval of mean absolute error (MAE) = 5.55 years (r = 0.91, R^2^ = 0.82). The top 10 biomarkers included albumin, erythrocytes, glucose, alkaline phosphatase, hematocrit, urea, RDW, cholesterol, alpha-2-globulin, and lymphocytes. Mamoshina P. et al. [24] presented a new aging clock trained on the data from several populations. The most effective predictor achieved an MAE of 5.94 years despite being trained on fewer features (21 vs. 41). It is likely that ethnically diverse aging clocks are more accurate than conventional ones in predicting chronological age and measuring biological age. The most important blood biochemistry parameters for all three populations were albumin, glucose, urea, and hemoglobin.

The models laid the foundation for the following calculators: Aging.AI 1.0 (r = 0.91, Rsq = 0.82, MAE = 5.5 years), Aging.AI 2.0 (r = 0.79, Rsq = 0.63, MAE = 6.2 years), and Aging.AI 3.0 (r = 0.8, Rsq = 0.65, MAE = 5.9 years) [25]. The predictors use various combinations of input parameters: albumin, glucose, alkaline phosphatase, urea, erythrocytes, cholesterol, RDW, alpha-2-globulins, hematocrit, alpha-amylase, lymphocytes, ESR, total and direct bilirubin, gamma GT, creatinine, LDH, total protein, alpha-1 globulins, beta globulins, gamma globulins, triglycerides, chlorides, HDL-C, LDL-C, calcium, potassium, sodium, iron, hemoglobin, MCH, MCHC, MCV, platelets, leukocytes, ALT, AST, basophils, eosinophils, monocytes, and neutrophils. The parameters are measured in whole blood, plasma or blood serum.

Several authors have used the above predictors in their studies. Cohen [26] used 10 biomarkers from Aging.AI (albumin, glucose, alkaline phosphatase, urea, erythrocytes, cholesterol, RDW, alpha-2 globulins, hematocrit, and lymphocytes) to predict chronological age in cohorts from the Women’s Health and Aging Study I &II (WHAS), the Baltimore Longitudinal Study on Aging (BLSA), Invecchiare in Chianti (InCHIANTI) and publicly available cross-sectional data from a representative sample of the American population from the National Health and Nutrition Examination Survey (NHANES). The performance in all four data sets was not as robust, with MAE ranging from 12.7 (NHANES) to 17.4 (BLSA). The authors excluded the possibility that the results were due to the use of 10 biomarkers rather than 41 and suggested that it could be caused by the absence of children in the cohorts and the differences in ethnic, socioeconomic, and environmental backgrounds. Overall, the results were consistent with those reported by Putin E. et al. [10] and showed the model’s tendency to underestimate the age of individuals over 70 years of age, i.e., it lacked discriminatory power in older age ranges.

Psychological status-based calculation of biological age using medical history and self-estimation of physiological and emotional states.

Currently, there are extremely few papers on psychological markers of aging. However, they deserve further investigation, particularly due to the non-invasive nature of the associated procedures. Repeatedly, biological aging has been shown to lead to cognitive decline. Diagnosed cognitive dysfunction is a predictor of unsuccessful aging and mortality; however, it has a low predictive power in younger people. Zhavoronkov et al. used deep neural networks (DNNs) to classify human behavior for biological age prediction [27]. They presented two new models, PsychoAge and SubjAge, which were similar to the aging clock. To predict chronological and subjective age, they trained the DNNs on a set of 50 modifiable behavioral features based on anonymous surveys of U.S. residents from the Midlife in the United States (MIDUS). After filtering and exclusion, the final dataset comprised 6071 samples. DNNs were able to accurately predict age, with MAE = 6.7 years for chronological age and MAE = 7.3 years for subjective age. Both PsychoAge and SubjAge have also been shown to be predictive of the risk of all-cause mortality. For both models, the top five important variables were related to sex life in the past 10 years, marital status, health limitations on vigorous activity, and intake of prescription blood pressure drugs. Headache frequency in the past 30 days ranked 5th in PsychoAge and 9th in SubjAge. Neuroticism, one of the five most commonly used personality traits, was the only one present among the top 25 features in PsychoAge. Openness and extraversion, another “big fiver”, were the only personality traits in SubjAge.

**Table 1 ijms-23-15103-t001:** Main characteristics of the blood biochemistry-based and psychological status-based calculators.

Study	Sample	Validation Strategy and Model Characteristics	Model Parameters	Comments
Putin E. et al., 2016 [10]	62,419 samples from the Eastern European population (90% of Russia)	Highly biased markers excluded. The dataset divided into training and test sets of 56,177 and 6242 samples, respectively. 40 deep neural networks (DNNs) trained on 56,177 blood test samples. Modular ensemble comprising 21 DNNs of varying depth, structure and optimization to predict human chronological age using a basic blood test. The full version is based on 41 blood chemistry parameters.	Best performance (DNN): 81.5% epsilon-accuracy r = 0.90, R^2^ = 0.80, MAE = 6.07 years within a 10-year frame. Overall performance (ensemble): 83.5% epsilon-accuracy r = 0.91, R^2^ = 0.82, MAE = 5.55 years.	5 most important markers for chronological age estimation identified: albumin, glucose, alkaline phosphatase, urea, and erythrocytes An algorithm developed based on a single source of clinical data.
Mamoshina P. et al., 2018 [24]	20,699 samples for the Canadian population, 65,760 samples for the South Korean population, and 55,920 samples for the Eastern European population	55,751 samples from the NHANES with blood test values used to measure the predictive power of the models	The best-performing predictor trained on the Eastern European population-specific dataset demonstrated an MAE of 6.25, an R^2^ of 0.69 Best performing predictors: Canadian population: MAE = 6.36 years, R^2^ = 0.52 South Korean population: MAE = 5.59, R^2^ = 0.49 Eastern European population: MAE = 6.25, R^2^ = 0.69	Population type identified as a major feature for age estimation in all 3 populations
Zhavoronkov A. et al., 2020 [27]	The final dataset contained 6071 participants (U.S. residents)	(1) Series of DNNs were trained based on data from anonymized questionnaire responses from MIDUS 1, MIDUS 2 and MIDUS Refresher longitudinal surveys (2) 50 most important features were selected to build the final models (modifiable factors) (3) Final models were trained with five-fold cross-validation (CV) using all MIDUS 1 samples. MIDUS 2 and MIDUS Refresher were used for model validation purposes	Best performance (DNN): MAE = 6.70 years and epsilon accuracy = 0.78 for PsychoAge; MAE = 7.32 years and epsilon accuracy = 0.74 for SubjAge Model validation (using other datasets): MAE = 7.18/7.73 years and epsilon accuracy = 0.73/0.70 for PsychoAge; MAE = 8.53/8.56 years and epsilon accuracy = 0.66/0.65 for SubjAge	Additionally, variables that remain highly important (top-25) across all age groups were defined. These variables form the psychological aging core.

### 2.2. Age Predictors Based on Molecular and Genetic Markers

In terms of current translational medicine, the distinction between predictors based on clinical and molecular-biological biomarkers is rather arbitrary. Here, we present some of the models that are most technically and technologically advanced but still feasible in routine clinical practice. We omitted the models that are accurate but require extensive invasive procedures and are not easily applicable in clinical practice [28].

#### 2.2.1. Transcriptome-Based Age Predictors

Peters M. et al. were among the first to successfully use transcriptome analysis to predict age [29]. They investigated 14,983 whole-blood samples from people of European ancestry. To calculate the “transcriptomic” age based on age-related differential gene expression, the authors used Illumina HumanHT-12 (v3/v4) and identified 1497 genes that produced highly correlated results in the discovery and replication stages. The R^2^-values for chronological age and predicted transcriptomic age were below 0.6; however, the average absolute difference between the predicted and chronological age was 7.8 years. A limitation of this study and similar studies is the use of bead chip arrays that only query 1.6% of all CpGs in the genome, and the characteristic background noise may complicate the reproducibility of results. Fleischer et al. developed a computational method based on linear discriminant analysis. They analyzed genome-wide RNA-seq profiles of human dermal fibroblasts from 133 individuals aged 1–94 years [30]. The algorithms produced R^2^ of 0.81 for the actual versus predicted age, a 4-year median error and a 7.7-year mean absolute error. They also predicted accelerated aging in progeria patients (patients with Hutchinson-Gilford progeria syndrome). The authors used genome-wide transcriptome analysis; however, they focused only on one cell type—skin fibroblasts, which limits the scope of the predictor: changes in the expression of many age-related genes seem to be tissue-specific, and only a limited number of genes have shown similar expression changes across tissues [31].

Ren X. et al. developed RNAAgeCalc, an age calculator based on transcriptional activity across 30 different tissues [32]. They used genome-wide and transcript-level gene expression data from 9662 samples available from the Genotype-Tissue Expression (GTEx) Program (V6 release). Tumor samples (n = 102) were omitted. Across all tissues, 1616 genes were identified as age-related. Transcriptional age acceleration was significantly correlated with mutation burden, mortality risk and cancer stage in several types of cancer from the TCGA database. Despite the above advantages, RNAAgeCalc produced rather high median errors for the predicted transcriptional age and chronological age (7–10 years, for most tissues).

Mayer D. et al. developed a binarized transcriptomic aging (BiT age) clock, which is currently among the most accurate transcriptome-based age predictors [33]. They processed 1020 publicly available RNA-seq samples for adult C. elegans, 900 of which were used to train and test the model. The transcriptome data were binarized to reduce noise: with a count per million above the median of the corresponding sample, the value of each gene was set to 1; otherwise, it was set to 0. BiT age does not require age discretization and allows assessment of the effect of single gene expression changes on the predicted age. To demonstrate the applicability of the novel method, the authors used the same human datasets as Fleischer et al.; however, binarization before calculating the elastic net regression significantly improved the results: R^2^ = 0.92; the Pearson correlation = 0.96 (*p* = 7.87e-73), the Spearman correlation = 0.96 (*p* = 9.31e-73); MAE = 6.63 years; MAD = 5.24 years; and RMSE = 8.41 years. The model also predicted that the patients with Hutchinson-Gilford progeria syndrome (HGPS) were significantly older. BiT age comprises 141 predictor genes, among which the forkhead transcription factor FOXO1—a regulator of the aging process in C. elegans and mammals—is positively correlated with age, which serves as further evidence of the evolutionary conservation of transcriptional mechanisms that regulate longevity [34].

Another promising approach to calculating biological age is examining circRNAs. In the study by Wang J. et al. [35], regression tree, with an MAE = 8.767 years (S.rho = 0.6983) and random forest regression (RFR) with an MAE of 9.126 years (S.rho = 0.660), outperformed five other models. However, RNA-based (circular and micro) predictors have so far used small samples (up to 100 people) and require significant improvements. The study did not report on any associations with age-related diseases; however, recent studies have shown that the number of circRNAs in the brain changes in older people and could be a major factor for neurodegenerative diseases, such as Alzheimer’s disease (AD) and Parkinson’s disease (PD) [36,37,38]. Haque et al. [39] investigated circRNA expression in the blood of an aging person and found that circFOXO3 and circEP300 were expressed differentially in one or several cell types, which could be interpreted as indirect evidence of circRNAs’ potential as an age marker.

#### 2.2.2. Predictors Based on the Peripheral Blood Proteome

Lehallier B. et al. developed a bioinformatics approach by analyzing venous plasma from 4263 healthy people aged 18–95 years [40]. Before processing, plasma was treated with ethylenediaminetetraacetic acid (EDTA). The authors used the SomaScan aptamer technology for high-precision proteomic analysis. They found a significant sex-related difference in 895 out of 1379 proteins that changed with age (q < 0.05). Lehallier et al. concluded that aging is a series of biologically motivated surges in plasma protein levels. The test in 1446 individuals provided a 0.97 Pearson correlation coefficient between the chronological age and predicted age. The authors also demonstrated that deviations from the plasma proteomic clock were correlated with clinical and functional changes (Table 2).

#### 2.2.3. Metabolome-Based Age Predictors

Van den Akker E. et al. used 56 serum biomarkers and proton nuclear magnetic resonance (1H-NMR) to build metaboAge, a metabolomics-based age predictor of an individual’s biological age. The predictor achieved a high correlation coefficient between the predicted and chronological age, with an average mean absolute error of 7.3 years and R^2^ = 0.654. MetaboAge also proved effective in predicting current and future cardiovascular and metabolic health and functionality in older individuals [41].

#### 2.2.4. Age Predictors Based on T-Cell DNA Rearrangements

With age, the number of episomal DNA molecules, or signal joint T-cell receptor (TCR) excision circles (sjTREC), declines in a log-linear fashion. This is a manifestation of a persistent thymus involution that starts soon after birth: the thymus transforms into adipose tissue and loses its function. Zubakov et al. used the sjTREC number as the only predictor in a linear regression model, which explained a large share of highly statistically significant total age variance (R^2^ = 0.835, *p* = 8.16 × 10^−215^, standard error of the estimation ± 8.9 years) [42].

#### 2.2.5. Microbiome-Based Age Predictors

Galkin F, et al. developed an aging clock by analyzing more than 4000 metagenomic profiles of people aged 18–90 years. Floro’clock (R^2^ = 0.5, Rsq = 0.3, MAE = 5.9 years) uses whole-genome sequences of the intestinal lumen microbiota [43].

Huang S. et al. assessed the accuracy of several age prediction models based on oral, gut, and skin microbiome samples. The prediction ability differed in three models (mean ± standard deviation): the skin microbiome, 3.8 ± 0.45 years; the oral microbiome, 4.5 ± 0.14 years; the gut microbiome, 11.5 ± 0.12 [44].

Several promising age prediction strategies, such as those based on the assessment of DNA damage [45], have not been tested in large cohorts and cannot be considered reliable methods of evaluating the pace of aging. Certain age prediction methods are no longer in use. For instance, telomere length is currently not viewed as a biomarker of human aging due to its hypervariability across human tissues [46].

Currently, the predictors based on molecular and genetic markers present a promising approach to biological age prediction. However, further research is needed for their clinical application.

**Table 2 ijms-23-15103-t002:** Main characteristics of the estimators based on molecular and genetic markers.

Reference	Methodology	Accuracy Reported	Sample Types	Number of Samples (Overall)	Age Range (Years)	Comments	Associations with Age-Related Conditions and Diseases
Fleischer J. et al. [30]	Linear regression10-fold cross-validation	MAE = 7.7 years MAE (median) = 4 years	Human dermal fibroblasts	133 (people) 10 Hutchinson-Gilford progeria syndrome patients	1–94 2–9	Predicts progeria patients as 15–24 years older than age-matched controls; hence, provides an accurate estimation of biological age The only method that predicts accelerated aging in HGPS patients	No associations reported
Van den Akker E. et al. [41]	Linear regression 5-Fold-cross-validation	R^2^ = 0.65 MAE = 7.3 years	Blood metabolome	25,000	-	Only a biological sample required; no additional metadata needed Participants with current metabolic syndrome or diabetes mellitus type 2 were estimated older than healthy counterparts	Cardiometabolic health; increased risk of hospitalization due to heart failure, cognitive decline and cardiovascular and all-cause mortality; in nonagenarians, lower instrumental activities of daily living and increased risk of all-cause mortality during 10 years of follow-up
Ren X. et al. [32]	Elastic net	Multiple	Multiple	9662	-	Transcriptional age is significantly impacted by race The first model to perform RNA-Seq-based identification of differential gene expression for each individual tissue type	Significant correlation between the transcriptional age acceleration and mutation burden, mortality risk, and cancer stage in several types of cancer; Complementary information to DNA methylation age
Meyer D. et al. [33]	Temporal scaling and binarization	R^2^ = 0.92 MAE = 6.63 years MAD = 5.24	Blood	1020	-	Universal applicability, no methylation analysis required; Improved accuracy for HGPS patients compared with Fleischer’s transcriptome-based model; no DNA methylation in C. elegans, hence the effect of the epigenetic clocks in gene expression is unclear.	No associations reported
Wang J. et al. [35]	Multivariate linear regression (MLR) Regression tree (best performing) Bagging regressionRandom forest regression (RFR, best performing) Support vector regression (SVR)	MAE = 8.767 years (S.rho = 0.6983) MAE = 9.126 years (S.rho = 0.660)	Blood	100	19–73	Significantly smaller prediction MAE values for males than females (MAE = 6.133 years for males and 10.923 years for females in the regression tree model)	No associations reported
Peters M. et al. [29]	Meta-analysis	Multiple	Blood	14,983 (individuals)	-	Lower predictive accuracy compared to epigenetic clocks	Higher systolic and diastolic blood pressure, total cholesterol, HDL cholesterol, fasting glucose levels and body mass index (BMI)
Zubakov D. et al. [42]	Linear regression with sjTREC as a single predictor	R^2^ = 0.835, *p* = 8.16 × 10^−215^ Standard error of the estimate ± 8.9 years;	Blood	195 (individuals)	0–80	Storage time analysis showed no statistically significant difference between the sjTREC quantifications in fresh and 1.5-year-old blood samples of the same individuals	No associations reported
Galkin F. et al. [43]	Elastic Net (EN) Random Forest (RF) Gradient Boosting (XGB) Deep Neural Networks (DNNs)	MAE = 5.91 years	Stool	4000	18–90	Accuracy comparable to the existing DNAm solutions (MAE < 5 years) The microbiome composition (such as Akkermansia muciniphila, a marker of obesity, glucose metabolism, and overall intestinal health) could be used in diagnosing gut metabolism disorders; further research needed due to inconsistent results	No associations reported
Lahallier B. et al. [40]	SomaScan assay	Multiple	Plasma	2925	18–95		At peaks 2 and 3 (at the ages of 60 and 78), the proteins were associated with cardiovascular diseases, as well as Alzheimer’s disease and Down syndrome

### 2.3. Epigenetic Clocks

Epigenetic clocks predict biological age based on DNA methylation levels (cytosine-5 methylation within CpG dinucleotides). The review and comparison of epigenetic clocks is presented in Table 3. 

The first generation of epigenetic clocks comprised a set of CpG sites and used chronological age as reference. Bocklandt S. et al. were the first to calculate biological age by measuring methylation in CpG loci [47]. They identified 88 loci in or near 80 age-correlated genes. Methylation in three sites had the highest correlation with age and the widest distribution of values. The findings were validated in a different cohort. The authors developed a regression model based only on loci located in EDARADD and NPTX2 (error = 5.2 years). In 2013, Hannum G. et al. developed an epigenetic clock based on 71 methylation markers and clinical parameters (gender and BMI). The Hannum’s model produced an error of 3.9 years for the primary cohort and 4.9 years for the validation cohort. Although the authors focused on white blood cells (WBC), the model proved applicable to other human tissue types [48]. Horvath S. et al. carried out a more comprehensive analysis of methylation and developed a multi-tissue predictor measuring methylation levels in various types of tissues, such as whole blood, peripheral blood WBC, umbilical cord blood, brain tissues, neurons and glial cells, buccal epithelium, gastrointestinal tract, heart, lungs, kidneys, saliva, placenta, etc. Using multivariate regression, the model automatically selected 353 methylation sites, which make up the predictor. It showed a fairly high accuracy on both the training set (age correlation = 0.97, error = 2.9 years) and test set (age correlation = 0.96, error = 3.6 years). The pace of aging in different tumor tissues was significantly accelerated (by an approximate average of 36 years), while the pluripotent stem cells had a DNAm age close to zero [49]. In 2014, Weidner et al. [50] developed a model based on 102 CpG methylation sites in blood. To facilitate clinical application, they focused on three methylation sites in the highly age-correlated genes—ITGA2B, ASPA, and PDE4C (training set, MAD = 5.4 years; validation set, MAD = 4.5 years). In 2018, Horvath S. et al. were able to improve their model [51]. They used different tissue; specifically, they increased the sensitivity for fibroblasts, since skin biopsy and isolation of fibroblasts are widely used in progeria research. The new age estimator comprised 391 CpGs. It has been used in several studies to calculate life expectancy or assess all-cause mortality [52,53,54] and analyze the association between biological age and aging-associated diseases [55,56,57,58]. Hun Y. et al. carried out a comparative analysis of biological age models based on various methods of measuring CpGs, such as 450 k Beadchip platform, CpG pyrosequencing, droplet digital PCR (ddPCR), and bisulfite barcoded amplicon sequencing. The ddPCR-based model was the most accurate in predicting epigenetic age in an independent validation sample, which could be due to the fact that the PCR is generally characterized by low error rates [59]. Galkin F. et al. used deep neural networks and data from 17 studies to develop an aging model comprising 1000 sites, with an MAE of 2.77 years [60].

Second-generation epigenetic clocks were the next step in the quest for more accurate and robust biomarkers of aging, including clinical characteristics. Researchers were not satisfied with the existing epigenetic clocks that used chronological age as a surrogate measure of biological age and did not factor in CpG sites, the methylation of which was not strongly associated with age. In 2018, Levine M. et al. developed a new epigenetic biomarker of aging—DNA-m PhenoAge [61]. First, they built a model that calculated phenotypic age based on nine clinical markers most significantly associated with mortality (albumin, creatinine, serum glucose, c-reactive protein, lymphocyte percentage, mean cell volume, red cell distribution width, and alkaline phosphatase) and chronological age. During the second stage, they selected 513 CpG sites that were the most accurate in predicting phenotypic age and that formed DNA-m PhenoAge. The model showed a significant correlation with all-cause mortality, age-related diseases, cardiovascular diseases, coronary artery disease, incidence of and mortality from lung cancer, Alzheimer’s disease, etc. Out of 513 CpGs, 41 CpGs were the same as in the Horvath DNAm age measure and six CpGs as in Hannum’s clock. In 2019, Lu A. et al. proposed a modified GrimAge model that was developed in two stages [62]. First, they identified DNAm biomarkers of physiological risk and stress factors (adrenomedullin, C-reactive protein, plasminogen activation inhibitor 1 (PAI-1), and growth differentiation factor 15 (GDF15)). They then combined them into one complex biomarker, DNAm GrimAge, and carried out a large-scale meta-analysis. The authors demonstrated that DNAm GrimAge was an accurate predictor of time-to-death, time-to-cancer, time-to-CVD, time-to-fatty liver, and time-to-menopause.

Thus, epigenetic clocks are among the most promising biomarkers of biological age and powerful predictors of lifespan. However, there are approximately 28 million CpGs in the human genome, and the above models only used approximately 20,000 CpGs available in 27 K, 450 K, and EPIC. Publicly available whole-genome bisulfite sequencing databases would greatly facilitate the development of even more accurate epigenetic clocks [63].

**Table 3 ijms-23-15103-t003:** Epigenetic clocks based solely on methylation sites.

**Epigenetic Clocks (Based Solely on Methylation Sites)**							
**Title**	**Sample**	**Aged**	**Biomaterial**	**Methods**	**Regression Model**	**Results**	**Model Parameters**
Bocklandt S. et al., 2011 [47]	128	21–55	Saliva	Microarray analysis: Illumina Human Methylation 27 microarrays Validation: Mass Array (Sequenom) and pyrosequencing	Multivariate regression and leave-one-out analysis	A total of 88 CpGs identified A linear model built based on 2 methylation sites in Edaradd и NPTX2	MAE for males only, 5.3 y. MAE for females only, 6.2 y. MAE combined, 5.2 y.
Hannum J. et al., 2013 [48]	656	19–101	Whole blood	Illumina Human Methylation 450 BeadChip assay.	Penalized multivariate regression method (Elastic Net) combined with a bootstrap approach	A linear model built based on 71 CpGs and included gender and BMI	Training data: R = 96%, RMSE = 3.9 y. Validation data: R = 91%, RMSE = 4.9 y.
Horvath S., 2013 [49]	7844 non-cancer 5826 cancer	0–100	Various human tissues and cell types	Illumina 27 K and Illumina 450 K platforms (for 21,369 CpGs present in both)	Penalized multivariate regression method (Elastic Net)	An aging clock formed by 353 CpGs automatically selected	Training data: R = 0.97, error = 2.9 y. Validation data: R = 0.96, error = 3.6 y.
Weidner C. et al., 2014 [50]	575	0–78	Whole blood	Human Methylation 27 BeadChip platform, Illumina Human Methylation 450 BeadChip assayBisulfite pyrosequencing for the 3 CpGs-based model	Multivariate linear regression	A predictive model developed by training on 102 CpGs and validated on 3 datasets and data from Hannum et al., covering 99 CpGs 3 CpGs selected by the multivariate linear model	Training data: MAD = 3.34 years, RMSE = 4.26 years, R^2^ = 0.98. Validation data: 3 datasets—MAD = 5.79, 5.52, and 4.02 years, respectively, Hannum et al. dataset—MAD = 4.12 years, RMSE = 5.34 years, R^2^ = 0.87. 3 CpG model: MAD = 5.4 years, RMSE = 7.2 years; validation, MAD = 4.5 years and RMSE = 5.6 years
Horvath S. et al., 2018 [51]	2222	0–92	Whole and cord blood, skin and buccal epithelium, fibroblasts	Infinium 450 K и EPIC array 850 K	ElasticNet regression	Epigenetic age estimator based on 391 CpGs.	Fibroblasts: R^2^ = 0.91, err = 2.6; Epithelium: R^2^ = 0.94, err = 6.3) Buccal cells: R^2^ = 0.88, err = 2); Keratinocytes: R^2^ = 0.99, err = 1); Skin: R^2^ = 0.99, err = 2.9
Han et al., 2020 [59]	973 40 38	1–101	Whole blood	450 K Illumina Bead Chip Pyrosequencing Droplet digital PCR Bisulfite barcoded amplicon sequencing	Linear correlation with the logarithm of age A multivariable linear regression model A multivariable model A multivariable linear regression model	65 CpGs-based model 6 CpGs-based models 7 CpGs-based models 9 CpGs-based models	Training set R^2^ = 0.95; MAE = 3.0 years; Validation on 3674 samples: R^2^ = 0.82; MAE = 3.3 years. Validation on 40 samples: R^2^ = 0.86; several months later median error = 6.8 years. Validation on 40 samples: R^2^ = 0.89; median error = 2.9 years. The training set (R^2^ = 0.95; median error = 2.8 years); validation on 39 samples: R^2^ = 0.87; median error = 2.4 years.
Galkin et al. [60]	6411	≈0–100	Whole blood	Infinium Human Methylation 450 K and 27 K BeadChip platforms	Deep neural network	1000 CpG-based model	MAE = 3.80 y. MedAE = 2.77 y. R^2^ = 0.93
**Combination Clocks (Epigenetic + Clinical Biomarkers)**							
Levine M. et al., 2018 [61]	456	21–100	Whole blood	Illumina 27 K and Illumina 450 K platforms, EPIC array 850 K (20,169 CpGs)	Elastic-net regression	513 CpGs-based model	Strong associations between DNAm PhenoAge and all-cause mortality, mortality from aging-related diseases, CVD and coronary heart disease (CHD) mortality, cancer incidence and mortality, and Alzheimer’s disease.
Lu A. et al., 2019 [62]	2356		Whole blood	Illumina 450 K platforms, EPIC array 850 K	Elastic-net regression	1030 CpGs-based model	Accurate prediction of time-to-death, time-to-cancer, time-to-CVD diseases, time-to-fatty liver, and time-to-menopause

## 3. Clinical Application of Biological Age Predictors

The clinical implications of biological age calculators cannot be overstated. Numerous studies have shown that accelerated biological aging is associated with a shorter lifespan, early menopause, the onset and progression of cardiovascular and metabolic diseases, fatty liver, cancer, etc. Age was identified as a primary determinant of the course of COVID-19 early in the pandemic; hence, establishing the patient’s age is ever more relevant for accurate prediction of the course of infection. Furthermore, it has been hypothesized that the effect of biological age on the course of the disease may be even greater than that of chronological age.

Galkin F. et al. used BloodAge, a deep learning aging clock, to calculate the pace of aging in 5315 COVID-19 patients. They found that the pace of aging was a stronger determinant of lethal outcome than chronological age [64].

Corley M. et al. used PhenoAge to evaluate the acceleration of epigenetic age and GrimAge to assess the risk of mortality in patients with severe COVID-19. Epigenetic age was much more accelerated in the patients with severe COVID-19 than in the controls and influenza patients. The DNA methylation analysis, however, showed no significant reduction in telomere length in the patients with severe COVID-19 [65]. Ying K. et al. assessed three different risk-based biological age predictors for UK Biobank subjects. Phenotypic Age and Dynamic Organism State Indicator (DOSI) provided 1.28- and 1.31 odds ratios of COVID-19 infection (95% CI: 1.25–1.31; *p* = 8.4 × 10^−82^; 95% CI: 1.26–1.38; *p* = 9.5 × 10^−32^, respectively) for every 10-year add-on to biological age [66]. Kuo C-L. et al. concluded that PhenoAge estimates were better predictors of COVID-19 severity than chronological age. After adjusting for current chronological age and pre-existing diseases or conditions, positive COVID-19 tests were associated with accelerated aging 10–14 years before the COVID-19 pandemic (OR = 1.15 for every 5-year acceleration, 95% CI: 1.08 to 1.21, *p* = 3.2×10^−6^) and all-cause mortality (OR = 1.25, for every 5-year acceleration, 95% CI: 1.09 to 1.44, *p* = 0.002) [67].

The impact of SARS-CoV-2 on epigenetic age has been another widely discussed topic. Pang A. et al. [68] used a novel principal component version of epigenetic clocks in longitudinal studies [69] to measure the epigenetic aging in non-hospitalized pre- and post-COVID-19 patients and healthy controls. In the post-COVID-19 patients aged over 50, they observed an average 2.1-year increase in PCPhenoAge estimates and an average increase of 0.84 years in PCGrimAge estimates. Under the age of 50, PCPhenoAge estimates were, on average, 2.06 years lower, while PCGrimAge showed no significant differences. Cao et al. reported accelerated epigenetic aging in COVID-19 patients, particularly in severe COVID-19 cases, irrespective of age. The findings were based on Hannum, PhenoAge, skinHorvath, GrimAge clocks, and DNAm TL. However, acceleration was observed during the initial and most critical phases of COVID-19, and the pace of epigenetic aging returned to normal during recovery [70].

## 4. Conclusions

Average life expectancy has increased over the past one hundred years. Today, more people live to be the middle- and oldest-old. However, this global aging trend entails a higher prevalence of aging-associated diseases—some of the major causes of disabilities and mortality. Although aging is natural, it impairs some of the most vital biological functions, which may lead to death. The pace of aging is both individual and multifactorial. Therefore, assessment of biological age as an indicator of overall health is crucial. Accurate and straightforward age assessment tools would aid clinicians in providing personalized care, improved estimates of the current health and health risks, and individualized prevention strategies.

Routine clinical assessments of the pace of aging must measure age-related changes and must be simple, accurate, noninvasive, and inexpensive. They must rely on modifiable criteria that could be used as therapeutic targets.

We believe that age calculators meet the above requirements for routine clinical practice. Age calculators based on clinical markers are optimal for health screening. They facilitate the identification of risk groups for accelerated aging and development of individualized prevention strategies. The reviewed models could be used in routine clinical practice in their current forms. However, they still should be tested in various populations.

Mixed-type calculators have emerged in the past few years. They combine clinical and epigenetic features and provide a more comprehensive and reliable assessment of the pace of aging. These calculators, however, are more expensive, which could impede their routine application. Many calculators are still being improved and tested. We believe that analysis of epigenetic changes may soon become widely available in routine clinical practice.

## Figures and Tables

**Figure 1 ijms-23-15103-f001:**
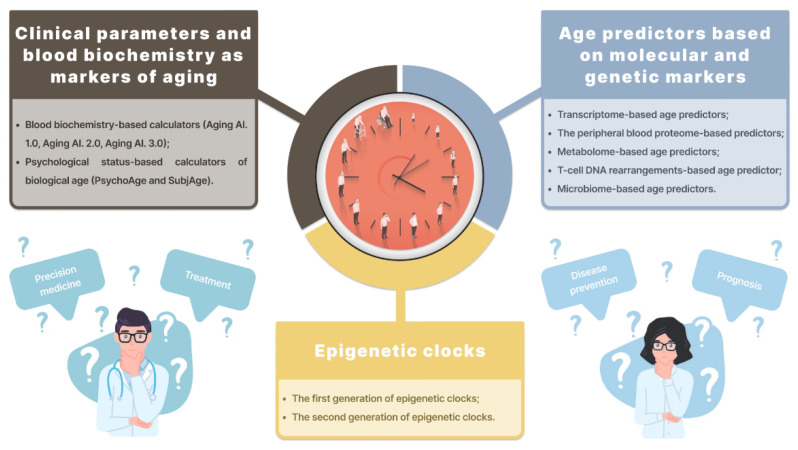
Biological age predictors mentioned in this review.

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
