# Peer review of "Biological Age Predictors: The Status Quo and Future Trends"

_ijms, 2022, doi:10.3390/ijms232315103_

Round 1
Reviewer 1 Report
This review paper summarized the findings of the current biological age predictor. It is comprehensive including genetic, epigenetic, and clinical markers. Although the studies are comprehensively described, critical assessment of the validation for each predictor needs to be listed. In addition, a figure that illustrates all the age predictors would be informative.
Author Response
Thank you ever so much for your suggestion. We have added visuals for the discussed biomarkers to provide the reader a quick reference point. This paper is not meant to be a critical review; hence, it does not follow any critical review algorithms, such as PRISMA. We sought to give detailed information on the currently available predictors/calculators, which, even in their current forms, could to be used in clinical practice.

Reviewer 2 Report
1. The section on "psychological biomarkers of age" needs some more evidence - it looks like it is only based off of one paper so far.
2. One question that should be discussed in the paper would be organ/tissue specific aging versus "whole-human" aging. Can different tissues/organs in the same individual have different rates of biological aging, and how should that be factored in while developing clinically relevant biomarkers of aging?
Author Response
Thank you for your review and valuable comments.
- Thank you ever so much for pointing this out. There are very few articles on psychological biomarkers, which are based on a sufficient sample size and validation cohort. We only reviewed articles that presented reliable and reproducible validation data. Hence, only one research made into the review.
- Indeed, the organs age at a different pace. We do believe that some of the organ-specific markers, such as lymphocytes, could be valuable markers of the overall aging. In clinical practice, however, we need more complex markers that more accurately reflect the aging of the body. Since this question require extensive discussion, we haven’t included it into this review.

Reviewer 3 Report
The work is thorough in most parts (although I feel the Covid-related chapters being somewhat over-represented, in contrast, e.g. West-Nile virus infection is not mentioned at all, although it has long been known to be equally age-related in outcome, well before the emergence of SARS2).
The perspective that I miss entirely, however, is the one dealing with the business / ethics opposition. The mentioned time-to-death, time-to-cancer, time-to-CVD, time-to-fatty liver etc. figures are highly sensitive from pharma / insurance / GDPR / ethical points of view, yet are totally missing from the manuscript. Adding such perspectives would provide a whole new layer to the overall story.
Author Response
Thank you very much for your comment. Our focus on COVID-19 reflects the current epidemiological situation. The disease persists, claiming lives, health and wellbeing of millions of people around the globe. Moreover, accelerated biological aging is a risk factor of severe Covid-19.
Indeed, pharma/insurance/GDPR/ethics are extremely important aspects. However, in our study, we focused solely on the review of clinical applicability of the age calculators and their performance characteristics. The above aspects warrant a separate, detailed investigation into the implications of age predictors, which should be based on solid facts and meticulous calculations and include economic and legal frameworks. Our aim was to provide a detailed, non-critical review of the status quo to facilitate clinical application. The paper is meant to highlight the available inventory of tools with their performance characteristics. Hence, the above aspects fall outside its scope. Moreover, we currently do not employ specialists required for a pharma/insurance/GDPR/ethical analysis, and our estimations and suggestions would be too broad to provide any meaningful contribution to the subject(s).
